# Post-exposure protection of SARS-CoV-2 lethal infected K18-hACE2 transgenic mice by neutralizing human monoclonal antibody

Ronit Rosenfeld [1 ✉], Tal Noy-Porat [1], Adva Mechaly[1], Efi Makdasi[1], Yinon Levy[1], Ron Alcalay[1], Reut Falach[1], Moshe Aftalion[1], Eyal Epstein[1], David Gur[1], Theodor Chitlaru[1], Einat B. Vitner [1], Sharon Melamed[1], Boaz Politi[1], Ayelet Zauberman[1], Shirley Lazar[1], Adi Beth-Din[1], Yentl Evgy[1], Shmuel Yitzhaki[1], Shmuel C. Shapira[1], Tomer Israely [1] & Ohad Mazor [1 ✉]

The coronavirus disease 2019 (COVID-19) pandemic, caused by the severe acute respiratory syndrome coronavirus 2 (SARS-CoV-2), exhibits high levels of mortality and morbidity and has dramatic consequences on human life, sociality and global economy. Neutralizing antibodies constitute a highly promising approach for treating and preventing infection by this novel pathogen. In the present study, we characterize and further evaluate the recently identified human monoclonal MD65 antibody for its ability to provide protection against a lethal SARS-CoV-2 infection of K18-hACE2 transgenic mice. Eighty percent of the untreated mice succumbed 6–9 days post-infection, while administration of the MD65 antibody as late as 3 days after exposure rescued all infected animals. In addition, the efficiency of the treatment is supported by prevention of morbidity and ablation of the load of infective virions in the lungs of treated animals. The data demonstrate the therapeutic value of human monoclonal antibodies as a life-saving treatment for severe COVID-19 infection.

[1] Israel Institute for Biological Research, Ness-Ziona, Israel. ✉email: ronitr@iibr.gov.il; ohadm@iibr.gov.il

Human monoclonal antibodies (mAbs), specifically targeting surface viral proteins, have increasingly demonstrated prophylactic and therapeutic efficacy against various viruses, including HIV, Ebola, and the pathogenic beta-coronaviruses Middle East respiratory syndrome coronavirus and severe acute respiratory syndrome coronavirus (SARS-CoV)[1–5]. Neutralizing Abs constitute a highly promising approach for treating and preventing infection by the novel SARS-CoV-2 (ref. [6]). The viral surface spike glycoprotein is essential for viral attachment, fusion, and entry into human cells, and thus considered as the major target for therapeutic neutralizing Abs[7–13]. Specifically, highly potent neutralizing Abs target and block the binding of the receptor binding domain (RBD) located in the S1 subunit of the spike to the human angiotensin-converting enzyme 2 (hACE2)[7,10,14].

A reliable animal model for COVID-19 is essential for the development of anti-SARS-CoV-2 countermeasures and for deciphering the pathogenicity of the disease[15]. Accordingly, mouse models that exploit the recombinant hACE2 expression, either by transgenic or by viral-transduction approaches were developed[16–28]. A transgenic mouse strain expressing hACE2 under the K18 promoter (K18-hACE2) was shown to be highly susceptible to SARS-CoV-2 infection, resulting in significant viral load in the lungs, heart, brain, and spleen as well as mortality[26,27,29]. In response to the urgent need for Ab-based therapy for COVID-19, several reports have demonstrated efficacy against SARS-CoV-2 infection by neutralizing Abs, primarily as prophylactic protection[10,19,30–32]. These studies were based on non-lethal COVID-19 models of mice transduced to express hACE2, and did not demonstrate the efficacy of Ab-based passive therapy administered at a significant time post-infection.

We have previously reported the isolation of human-neutralizing Abs selected against SARS-CoV-2 RBD by extensive screening of a phage-display library generated from lymphocytes collected from infected individuals. Among these Abs, MD65 exhibited the highest neutralization potency in vitro[11].

In this work, we evaluate the therapeutic efficacy of the MD65 Ab in a lethal COVID-19 animal model by assessing its prophylaxis and treatment abilities to protect K18-hACE2-infected mice.

## Results and discussion
### Reformatting and characterization of the MD65 Ab.
The initial binding and neutralization characterization of MD65 Ab were conducted following the expression of this Ab as a single-chain human-Fc recombinant form (scFv-Fc)[11]. Here, towards implementation of this Ab as a bona fide therapeutic product appropriate for human use, it was re-cloned and produced as a full recombinant IgG molecule of the IgG1/k isotype which includes the triple mutation M252Y/S254T/T256E (YTE) in the Fc region. These modifications increase the Ab affinity towards the human FcRn[33] at acidic pH and thereby prolong its serum half-life, a parameter that is essential for a high therapeutic value[34–36]. Characterization of three versions of the MD65 Ab: scFv-Fc, IgG, and IgG-YTE, established that overall they are comparable with regard to their antigen-binding performance (Fig. 1a) and that the IgG versions exhibit slightly favorable affinity when compared to the scFv-Fc one (apparent $K_D$ of 0.7 versus 1.0 nM, respectively). Similarly, the two MD65 IgG formats possess equivalent SARS-CoV-2 neutralization potency in vitro ($NT_{50}$ of 40 ng/ml), which is slightly better than that of the scFv-Fc Ab ($NT_{50}$ of 67 ng/ml; Fig. 1b). In addition, all three Ab versions were shown to effectively prevent the binding of SARS-CoV-2 spike RBD to the hACE2, again with favorable kinetics for the IgG formats (Fig. 1c). These slight differences may indicate that there is a correlation between Ab affinity and SARS-CoV-2 neutralization. This aspect will be further clarified in future studies.

### Pharmacokinetics of MD65 IgG-YTE.
A pre-requisite for achieving long circulatory half-life of a given Ab in humans, is to improve its interaction with the FcRn at low pH. We therefore verified that MD65 in its IgG-YTE format, as anticipated, exhibits improved pharmacokinetic abilities compared to the WT IgG format. Accordingly, the interactions of MD65 IgG and IgG-YTE with human FcRn, at pH 6.0, were assessed by biolayer interferometry (BLI). As expected, the modified Ab demonstrated a marked increase in affinity compared to the non-modified version, manifested by a threefold improved on-rate and significant lower off-rate of binding (Fig. 2a, b). Steady-state analysis of these interactions revealed that the overall affinity of the IgG-YTE version toward human FcRn at low pH is 10-fold higher than the IgG version (Fig. 2c).

As part of the pre-clinical evaluation of MD65 IgG-YTE Ab and prior to the examination of its therapeutic potency, the pharmacokinetic profiles following intravenous (IV; Fig. 2d) or intraperitoneal (IP; Fig. 2e) administration were determined. Accordingly, MD65 IgG-YTE was first intravenously injected to C57BL/6 mice and plasma Ab levels at various time points were determined by enzyme-linked immunosorbent assay (ELISA). The data were fitted by non-compartmental analysis, establishing that MD65 IgG-YTE exhibits a biphasic elimination profile

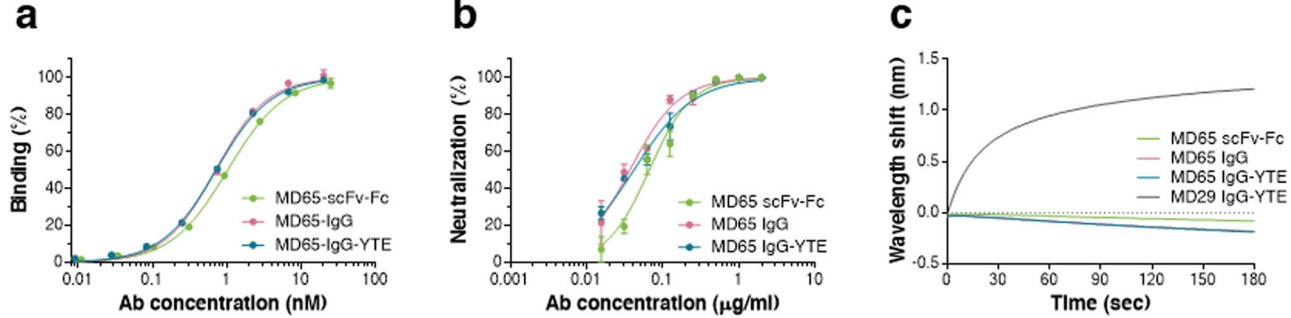

**Fig. 1 Characterization of the MD65 Ab versions. a** Binding profiles of MD65 Ab variants (specified by different colors as indicated in the respective inset legends) tested by ELISA against S1. Values along the curve depict averages of triplicates ± SEM. **b** SARS-CoV-2 in vitro neutralization potency of the MD65 Ab variants evaluated by plaque reduction neutralization test (PRNT). Values are averages of triplicates ± SEM. **c** BLI-determined binding of hACE2 to RBD in the presence of MD65 Ab variants or MD29 IgG-YTE (as a control). Each of the biotinylated antibodies was immobilized on a streptavidin sensor, saturated with RBD, washed, and incubated with recombinant hACE2 for 180 s. Time 0 represents the binding of the hACE2 to the antibody–RBD complex. The MD65 IgG curve completely overlapped with and is therefore masked by the MD65 IgG-YTE.

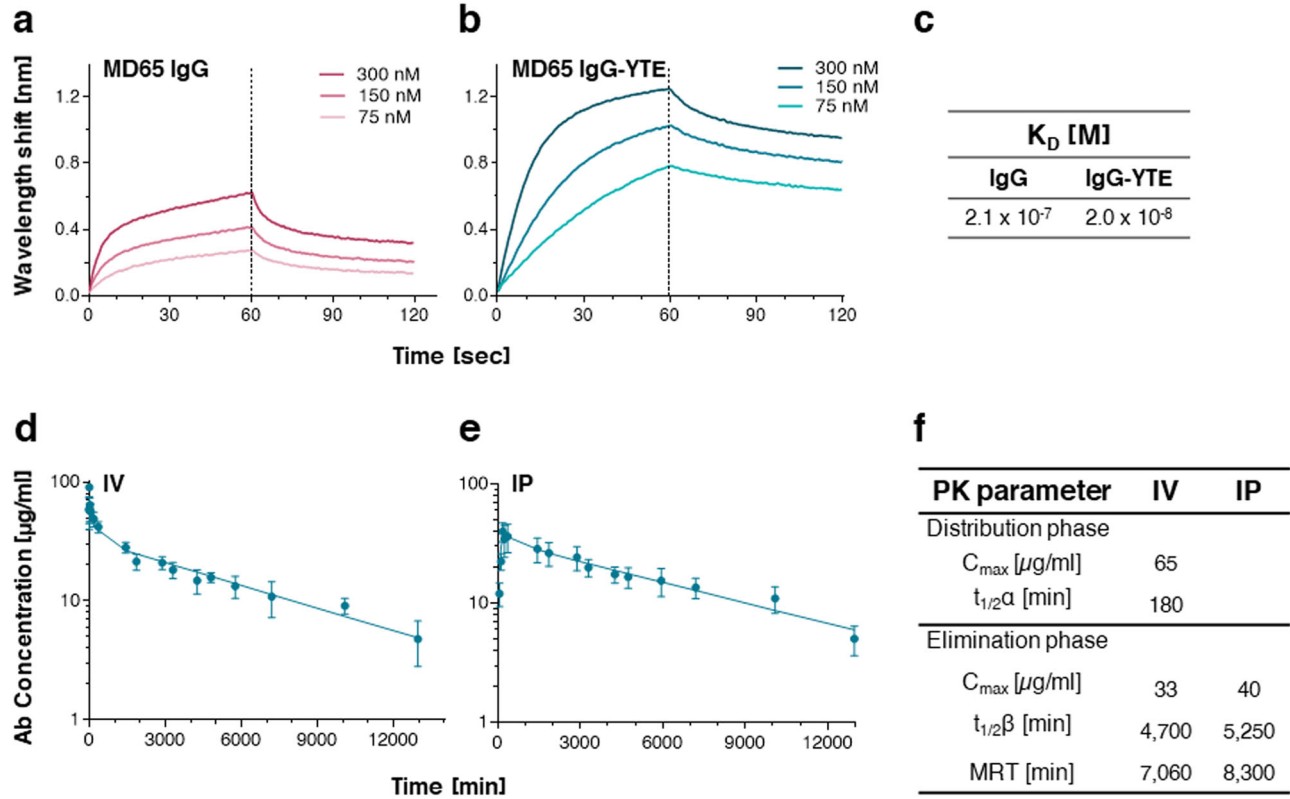

**Fig. 2 MD65 binding to human FcRn and pharmacokinetic analysis. a, b** Kinetics of the interactions between human FcRn and MD65 variants (**a** IgG, curves in magenta nuances; **b** IgG-YTE, curves in turquoise nuances) at pH 6.0, using BLI. Immobilized antibodies were interacted with the indicated concentrations of human FcRn for 60 s (association phase) followed by a wash step (dissociation phase). **c** Steady-state analysis of sensorgrams by 1:1 binding model was used to determine equilibrium $K_D$. **d, e** Plasma MD65 IgG-YTE concentrations in C57BL/6 mice were determined at the indicated time points following a single **d** IV ($n = 4$) or **e** IP ($n = 5$) administration of 0.2 mg Ab. Values are expressed as mean ± SEM that were fitted using non-compartmental analysis. **f** Pharmacokinetic parameters of MD65 IgG-YTE.

(Fig. 2d), consisting of a typical relatively short distribution phase ($\alpha$) with a $t_{1/2}$ value of about 180 min and a significantly longer elimination phase ($\beta$) with a $t_{1/2}$ value of about 4700 min (3.3 days; Fig. 2f). These results are in good agreement with our previous observation that a chimeric protein bearing the same Fc (non-YTE) exhibited similar pharmacokinetic parameters in mice[37], corroborating the notion that these mutations do not affect the Ab interaction with the murine FcRn.

The plasma levels of MD65 IgG-YTE following IP administration was then evaluated. The Ab exhibited a rapid and high bioavailability, reaching $C_{max}$ of 40 μg/ml within 180 min (Fig. 2e), a concentration which is commensurate to that measured in the sera at the same time point following IV administration (49 μg/ml; Fig. 2d). The elimination phase $t_{1/2}$ value following IP administration was approximately 5300 min (3.7 days; Fig. 2f). Therefore, the pharmacokinetic evaluation of the two alternative routes of administration indicate that they both are appropriate for the subsequent protection studies.

**Protection against SARS-CoV-2 lethal infection in K18-hACE2 transgenic mice.** To directly assess the use of the MD65 Ab as the basis for pre- and post-exposure therapy to countermeasure the SARS-CoV-2 infection, the K18-hACE2 transgenic murine model (derived from the C57BL/6 strain) was employed for in vivo-neutralizing studies. This model was recently shown to fatefully recapitulate the SARS-CoV-2 infection and consequently to serve as a reliable model to predict the efficacy of therapeutic strategies[18,26,27,38].

Infection of K18-hACE2 mice with 200 PFU of the SARS-CoV-2 BavPat1/2020 strain resulted in significant weight loss from day 5 after infection and death of 80% of the animals by day 9 post-infection (Fig. 3). Of note, experiments were conducted both with male and female mice establishing that no sex-dependent significant differences in weight loss or mortality occurred (as exemplified in Supplementary Fig. 1), in line with previous observations[29].

In order to maintain a steady Ab levels in the circulation for about 8 days (commensurate with the 7.5 days mean time to death (MTTD) in the control group), and taking into consideration the pharmacokinetic profile, the Ab was administered via the IP route in two successive treatments, 4 days apart. In sharp contrast to mock-treated mice (administered with phosphate-buffered salne (PBS) only), pre-exposure IP administration of the MD65 Ab 4 h before infection (first dose) resulted in full protection of the infected animals without any signs of weight loss (Fig. 3a–c) or any other clinical symptoms. The high lethality of the infection could not be alleviated in mice administered with an isotype antibody control (MH75) that contains the same Fc but targets a non-relevant antigen (ricin)[39].

The successful pre-exposure treatment of the infected mice prompted us to further evaluate the possibility to initiate treatment at even later time points. K18-hACE2 mice were therefore infected with SARS-CoV-2 and IP administered (first dose) with the MD65 Ab 1, 2, 3, or 4 days post-infection (dpi). It was found that treatment of infected mice initiated as late as 2 days after infection, was highly efficient in blocking the disease, as demonstrated by the complete prevention of weight loss and,

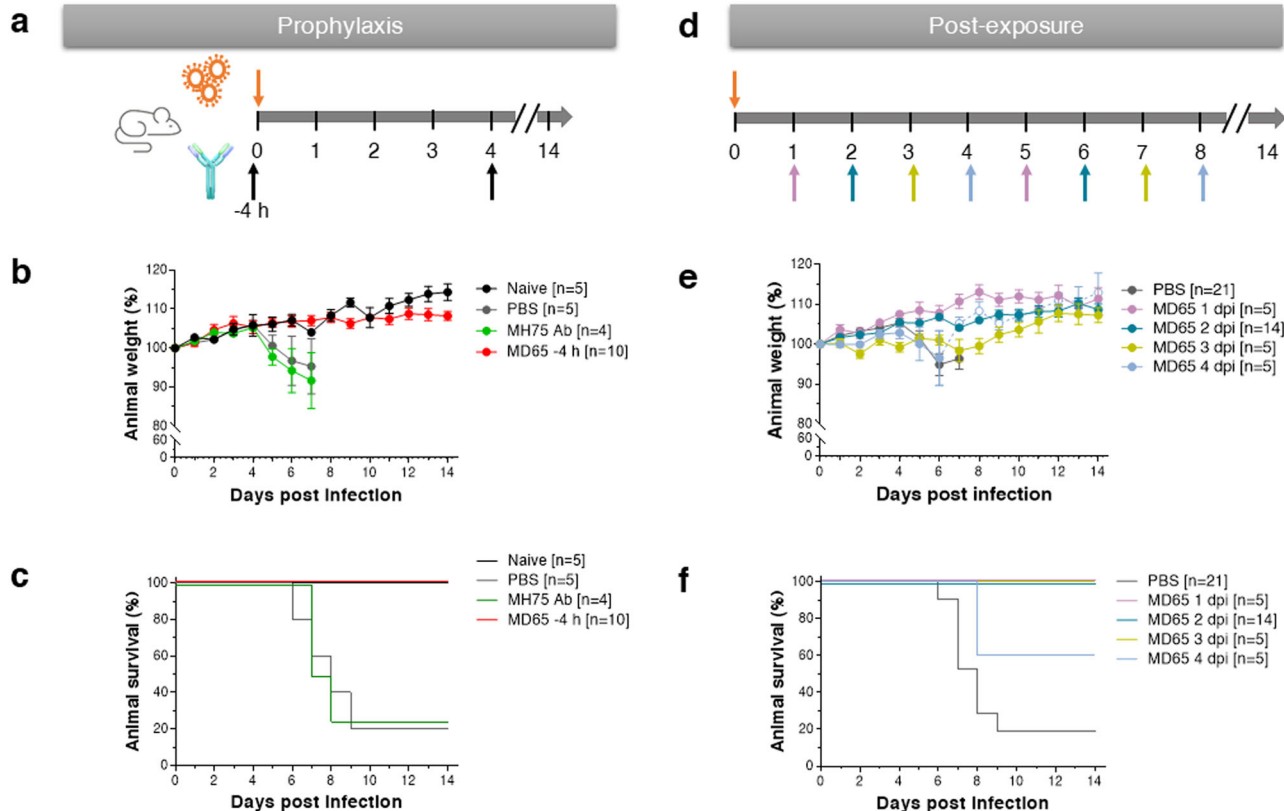

**Fig. 3 MD65 Ab-mediated prophylactic and post-exposure protection against SARS-CoV-2-infected K18-hACE2 mice.** Prophylactic (**a–c**) or post-exposure (**d–f**) in vivo protection experiments. Animals were intranasal infected with 200 PFU of SARS-CoV-2 BavPat1/2020 strain and IP administered with 1 mg/mouse of MD65 Ab at the indicated time points and 4 days later, for a second time. **a** Schematic description of the experimental design of the prophylactic treatment. **b** Body weight profiles. **c** Kaplan–Meyer surviving curves. Curves describe mice treated with the MD65 antibody (red line), naïve untreated and un-infected animals (black line), animals administered with PBS (gray line) or the isotype control irrelevant-Ab (anti-ricin MH75; green line) at the same time point as the MD65 treatment. **d** Schematic description of the experimental design of the post-exposure treatment. **e** Body weight profiles. **f** Kaplan–Meyer surviving curves. Curves describe mice treated with the MD65 antibody at various time points post-infection, as indicated by different colors in the legend within the panel and by the similarly colored arrows in panel **d**. Control animals (gray curves) were administered with PBS at day 2 and day 6 after infection. Body weight change is displayed as the percentage of initial weight. Only data of the first 7 days are presented in the control groups exhibiting significant mortality. In case of the mice treated 4 dpi (panel **e**), the weight of the surviving animals is indicated by hollow circles and dashed line. Data represent means ± SEM. The *n* numbers indicate the number of animals within each experimental group; when *n* > 5, the results were combined from several experiments in which each group included at least four animals.

most notably, by total protection of the mice (Fig. 3d–f). Furthermore, even when treatment was initiated 3 dpi, all treated animals survived. Yet, mice in this experimental group displayed some mild symptoms as manifested by a moderate and transient body weight loss. Even by further delaying the treatment to 4 dpi, a protection rate of 60% of the infected animals was observed, yet, without a significant delay in the time to death (for the 40% non-surviving animals). The surviving animals in this group regained their initial body weight (Fig. 3e), indicative of efficient recovery.

Taken together, these results demonstrate, for the first time, the therapeutic value of human mAbs as a life-saving treatment of severe and lethal COVID-19 infection model.

**SARS-CoV-2 viral load in lungs collected from infected K18-hACE2 mice.** The efficiency of the treatment with the MD65 was further established by directly interrogating the viral pulmonary load in infected mice, treated with the antibody 2 dpi (Fig. 4). The lungs were collected 6 dpi (the latest time point before the onset of mortality) and subjected to quantitative reverse transcriptase PCR (qRT-PCR) for quantification of the viral burden. The qRT-PCR data indicate that in the control non-treated group, as expected, the SARS-CoV-2 virus propagated and reached a very

high concentration (Fig. 4a), while Ab treatment resulted in a viral load at least one order of magnitude lower (*p* < 0.05).

We further evaluated the infectivity of the virus in the lung by testing its ability to infect VERO E6 cells and to form plaques. As expected, the virus that was isolated from the control group was viable and highly infective (Fig. 4b). However, no infective virus could be isolated from the lungs of the Ab-treated mice (Fig. 4b and Supplementary Fig. 2) establishing that MD65 treatment results in rapid and complete resolution of infection, in agreement with prevention of animal death. This latter finding also suggests that the viral load detected by PCR represents viral debris genomes present in the lungs and not bona fide infective virions.

**Lung histopathological evaluation of Ab-treated K18-hACE2 mice.** SARS-CoV-2 infection of K18-hACE2 mice is manifested by a progressive and widespread viral pneumonia with perivascular and pan-alveolar inflammation, immune cell infiltration, edema, lung consolidation, and distinctive vascular system injury that were apparent even 3 weeks after infection (in surviving animals)[26,29,40]. Here, the observation that post-exposure Ab-based treatment of infected mice resulted in complete protection

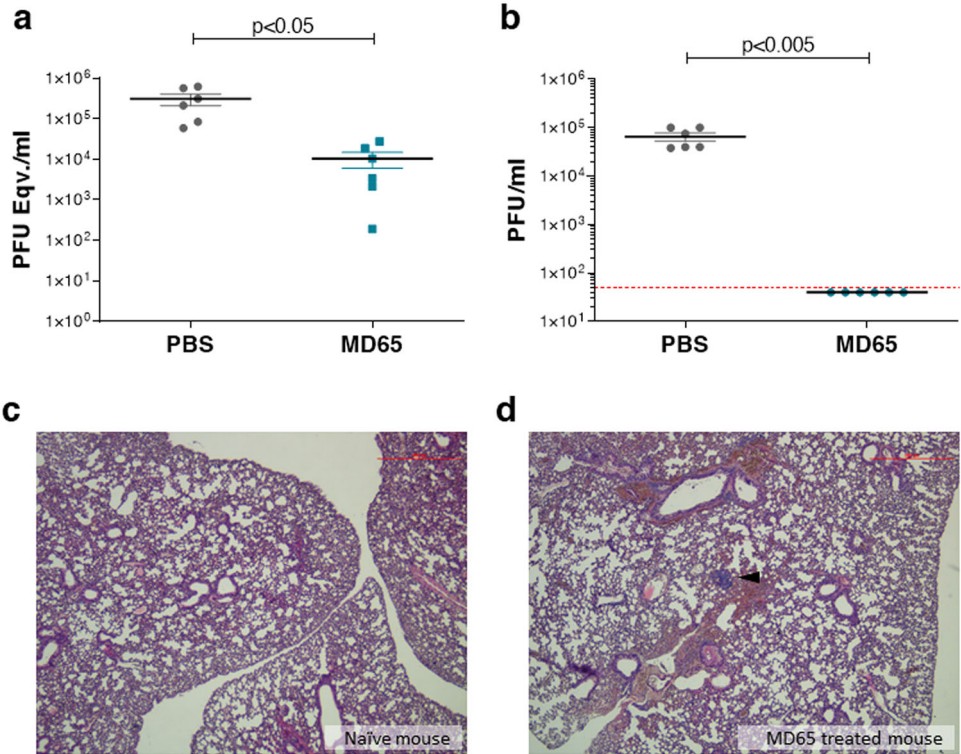

**Fig. 4 Viral load determination and histological analysis of lungs, collected from K18-hACE2 mice, infected with SARS-CoV-2, and treated 2 dpi with MD65 antibody. a**, **b** Viral load in lung samples, collected 6 dpi from PBS-treated mice ($n = 6$, gray dots) and mice treated 2 dpi with 1 mg MD65 antibody ($n = 6$, turquoise dots). **a** Viral load was quantified by qRT-PCR and expressed as equivalents of PFU/ml. **b** Infectious viral load determined by plaque assay. Dotted red line indicates the assay limit of detection. Data in **a**, **b** represent individual values and mean ± SEM. Horizontal bars indicate statistical significance of paired values. *P* values calculated using two-tailed paired *t*-test were: in panel **a**—0.0295 and in panel **b**—0.0031. **c**, **d** Histological analysis of lung sections, collected 21 dpi from a naïve mouse (**c**) and a mouse that was infected with SARS-CoV-2 and treated 2 dpi with 1 mg MD65 antibody (**d**); black arrow indicates lymphoid aggregate. The panels include representative images ($n = 2$ for the naïve mice and $n = 6$ for the MD65-treated mice, which were independently infected and analyzed, yielding similar results; full images of the lungs are provided in Supplementary Fig. 3). Magnification = ×40.

with no apparent signs of disease has prompted characterization of the pathological status of their lungs. To this end, mice were infected with SARS-CoV-2, treated with MD65 2 days later, and necropsy was performed at day 21. Overall, no major pathological changes or signs of inflammation were found in the lungs of the Ab-treated mice (Fig. 4c, d and Supplementary Fig. 3). The only indication of previous viral infection is the presence of scarce lymphoid aggregates that are well contained (Fig. 4d). It was previously reported that in the course of SARS-CoV-2 infection in the K18-hACE2 model, B cells infiltrated into the lungs, accumulated, and formed aggregates[29]. It is therefore conceivable to assume that the lymphoid aggregates observed here contain B cells that were recruited during the initial accumulation of the virus in the lungs, within the first days after infection. Thus, the effective Ab treatment that inhibited the virus ability to propagate have also prevented the progression of the inflammatory process, as strongly suggested by the infrequent occurrence of pathological changes in the lung and the alveolar tissue. These results substantiate the notion that post-exposure Ab treatment of lethal SARS-CoV-2 infection is highly effective, rapidly neutralizes the virus propagation, and limits the inflammatory and the pathological progression.

**Anti-SARS-CoV-2 seroconversion in Ab-treated animals.** Abs against the SARS-CoV-2 spike protein were reported to be detected as early as 7 days post-infection of K18-hACE2 mice[29]. It was previously shown that also following passive immunization (against other infections), surviving animals seroconverted by developing antigen-specific Abs, which contributed to protection

against re-infection[41]. Furthermore, in the context of the current pandemic, since the re-infection by SARS-CoV-2 is a tangible possibility, it was of interest to determine whether the MD65-treated mice developed an endogenous immune response toward the virus.

To this objective, the mouse humoral immune response towards SARS-CoV-2 Spike glycoprotein was measured in treated mice at 14 dpi (Fig. 5). Interestingly, the prophylactic administration of the MD65 Ab prevented the mounting of the humoral response, suggesting that effective neutralization of the virus occurred immediately upon infection (Fig. 5a). In contrast, animals treated either 1 or 2 days post-infection developed a marked Ab response that is in good correlation with the time of treatment. As several studies have shown that most SARS-CoV-2-neutralizing Abs target the S1 subunit and even more specifically the RBD, the humoral responses of the Ab-treated mice toward these two antigens was further evaluated. Indeed, a similar pattern was observed for these two antigens as for the spike (Fig. 5b, c), suggesting that the endogenous Ab response may contain SARS-CoV-2-neutralizing Abs that will confer partial or full protection against re-infection. Determination of the neutralization titer of these samples could provide further support for this concept, yet such an experiment is complicated by the masking effect of residual MD65 Ab.

**Dose-dependent therapeutic efficacy of MD65.** The ability of MD65 to protect K18-hACE2 mice from SARS-CoV-2 infection at lower single doses was further evaluated. To this end, single treatments of either 1, 0.1, and 0.01 mg Ab/animal were administered at

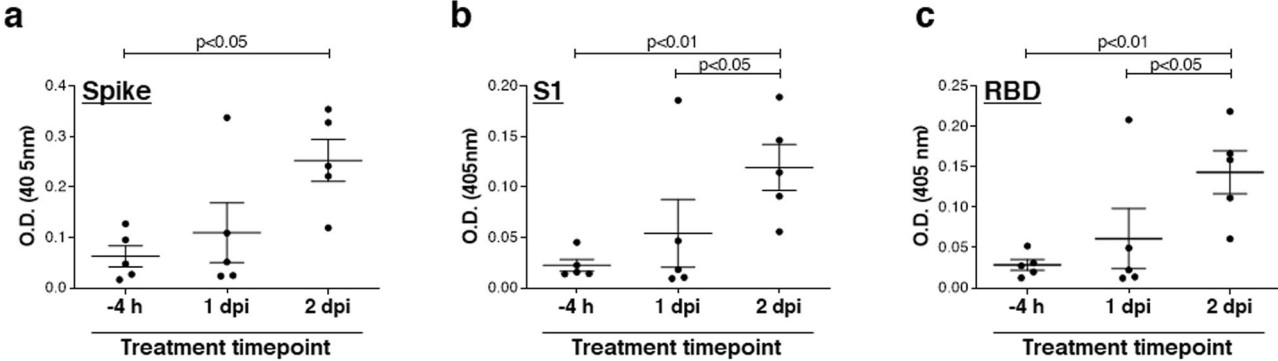

**Fig. 5 Endogenous anti-SARS-CoV-2 humoral response of infected K18-hACE2 mice treated with the MD65 antibody.** Sera samples were collected at day 14 post-infection from mice treated with MD65 at the indicated time points and tested by ELISA for the presence of endogenous (murine) antibodies against the SARS-CoV-2 antigens: **a** spike, **b** S1, or **c** RBD. Data represent individual and mean ± SEM ($n = 5$ mice) of measurements obtained for sera samples at 1:60 dilution. Horizontal bars indicate statistical significance of paired values. *P* values calculated using two-tailed paired *t*-test were: in panel **a** 0.0241; in panel **b** upper horizontal bar—0.0067, lower horizontal bar—0.0426; in panel **c** upper horizontal bar—0.0086, lower horizontal bar—0.0324.

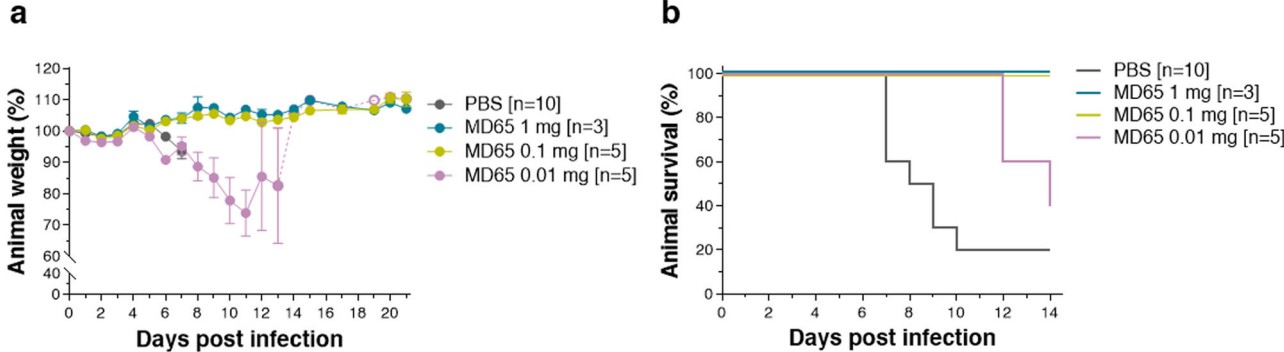

**Fig. 6 Dose-dependent therapeutic efficacy of MD65.** Single doses of 1, 0.1, and 0.01 mg MD65 Ab/animal were administered at day 2 post viral infection. **a** Body weight profiles. **b** Kaplan–Meyer surviving curves. Curves describe mice treated with the indicated doses of MD65 (see color legend within panels) and control animals, administered with PBS (gray line). Data represent means ± SEM. Body weight change is displayed as the percentage of initial weight. Only data of the first 7 days are presented in the control group exhibiting significant mortality. In case of the mice treated with 0.01 mg MD65, the weight of the surviving animal is indicated by hollow circles and dashed line.

day 2 post virus infection. Body weight loss or mortality were not detected in any of the animals treated with doses of 1 or 0.1 mg MD65 (Fig. 6). Conversely, no significant protection was attained by treating animals with the low 0.01 mg MD65 dose. In the latter case, 80% of the mice exhibited massive weight lost and succumbed to the infection. Yet, it should be noted that this dose of Ab induced a marked delay in the time to death (range 12–14 days with MTTD of 12.5 days) when compared to the control mice (range 7–10 days with MTTD of 8 days).

In conclusion, the dose-dependency experiments established that administration of MD65 as late as 2 dpi, at a dose as low as 0.1 mg/mouse (representing 5 mg/kg body weight) is therapeutically efficient. Furthermore, a therapeutically relevant delay in MTTD is promoted even at a 10 times lower dose.

Antibody-dependent enhancement (ADE) of infection was documented for several viruses including respiratory syncytial virus, measles, as well as for coronaviruses[42]. This phenomenon is usually associated with the enhancement of virus uptake by phagocytic cells or by the formation of immune complexes that lead to excessive inflammatory responses. Post-exposure treatment of infected animals, where the possibility for the formation of immune complexes is higher, was proposed as one of the tests for possible ADE effects in general and of SARS-CoV-2 in particular[42]. As of today, ADE was not reported for the current pandemic, yet there is a need to continue monitoring

for such effect[43]. In the current study, no worsened effect (in terms of mortality, body weight loss, as well as increased viral burden in the lungs) in the Ab-treated animals was observed, in spite of the high sensitivity of the K18-hACE2 model which could have detected with ease such an occurrence.

To conclude, the data presented in this report demonstrate high-effective post-exposure therapy of SARS-CoV-2 using a fully human mAb in a lethal infection animal model. While efficient therapeutic use of human mAbs has been previously succeeded[12,32], the current report documents efficient protection by administration of therapeutic Abs at significant late time points post-infection. From the clinical perspective, the efficacy of this Ab extends the therapeutic window permitting initiation of life-saving treatment at late stages post-infection. This study incents the development of a clinical product based on the MD65 Ab for human use.

## Methods
**Recombinant proteins expression.** Mammalian cell codon optimized nucleic sequence, coding for the SARS-CoV-2 Spike glycoprotein [GenPept: QHD43416 ORF (https://www.ncbi.nlm.nih.gov/protein/1791269090)], was used to design pcDNA3.1+-based expression plasmids, mediating recombinant expression of the soluble spike (amino acids 1–1207), S1 (amino acids 1–685), and RBD (amino acids 1–15 and 318–542) proteins. Same expression vector utilized for cloning and production of the secreted form of the hACE2 receptor [amino acids 1–740; GenPept: NP_068576.1 ORF (https://www.ncbi.nlm.nih.gov/protein/NP_068576.1)].

C-terminal his-tag as well as a strep-tag were included in order to facilitate protein purification. Expression of the recombinant proteins was performed using Expi-CHO™ Expression system (Thermoscientific, USA, Cat# A29133) following purification using HisTrap™ (GE Healthcare, UK) and Strep-Tactin®XT (IBA, Germany).

MD65 scFv, derived from a phage-display library, was initially cloned as a scFv-Fc Ab, as recently described[11]. To generate a full-length IgG, the VH and VL fragments where re-cloned into pcDNA3.1-Heavy and pcDNA3.1-Kappa vectors, respectively. These vectors were designed to include IgG1 and Kappa chain constant domains, as well as the appropriate signal sequences for the production of complete heavy and light chains. The resulting Ab was further engineered to include three point mutations in its Fc region: M252Y/S254T/T256E (YTE mutations). These mutations were previously shown to increase Ab affinity to the human FcRn, at pH 6.0 and hence to improve the Ab's serum half-life in rhesus macaques and humans[33,35,36]. The Ab MD29 scFc-Fc, reported to target RBD and neutralize SARS-CoV-2 but not inhibit ACE2–RBD interaction[11], was similarly cloned and expressed as IgG-YTE.

Recombinant Abs (scFv-Fc, IgG, and IgG-YTE) were expressed using ExpiCHO™ Expression system (Thermoscientific, USA, Cat# A29133) and purified on HiTrap Protein-A column (GE healthcare, UK).

All purified proteins were sterile-filtered and stored in PBS. The integrity and purity of the recombinant expressed proteins were confirmed by SDS-PAGE analysis (Supplementary Fig. 4).

**ELISA.** Direct ELISA was performed against the SARS-CoV-2 S1 subunit (expressed and purified as recently described[11]) for binding evaluation and for Ab concentration determination. Maxisorp 96-well microtiter plates (Nunc, Roskilde, Denmark) were coated overnight with 1 μg/ml of S1 protein in NaHCO₃ buffer (50 mM, pH 9.6), washed, and blocked with PBST at room temperature for 1 h. Human Abs were visualized by AP-conjugated Donkey anti-human IgG (Jackson ImmunoResearch, USA, Cat# 709-055-149, lot 130049) used at 1:1000 and further developed with PNPP substrate (Sigma, Israel, Cat# N1891). For the quantification of MD65 Ab presence in plasma or sera samples, standard curve included purified MD65 IgG-YTE Ab at the concentration range of 2–3000 ng/ml (diluted in PBST). Correct ELISA-based quantification was confirmed by four non-immune sera samples (collected prior to Ab-administration) containing exogenously added known concentration of the Ab. Standard curves confirmed as above were calculated for each individual ELISA plate/data set.

The endogenous humoral response was tested in mice sera samples collected at 14 dpi, evaluated by ELISA against SARS-CoV-2 spike, S1 and RBD essentially as described above, using AP-conjugated Donkey anti-mouse IgG (H + L) minimal cross (Jackson ImmunoResearch, USA, Cat# 715-055-150, lot 142717) used at 1:2000.

**BLI analysis.** Binding studies were carried out using the Octet system (ForteBio, USA, Version 8.1, 2015) which measures BLI. All steps were performed at 30 °C with shaking at 1500 r.p.m. in a black 96-well plate containing 200 μl solution in each well. For hACE2 competitive binding assay, streptavidin-coated biosensors were loaded with biotinylated MD65 (10 μg/ml), as either scFv-Fc, IgG, or IgG-YTE Ab formats. Ab-loaded sensors were first incubated with RBD (12 μg/ml), washed, and incubated with hACE2 (20 μg/ml). MD29 Ab, previously shown to bind RBD without blocking its binding to hACE2 (ref. [11]), was used as a control.

For the evaluation of MD65 affinity to human FcRn, anti-Fab-coated sensors were loaded with MD65 IgG or IgG-YTE (30 μg/ml), to reach 2.5 nm wavelength shift, and then washed. Sensors were then incubated with different concentrations of FcRn (Sino Biological #CT009-H08H; ranging from 75 to 300 nM) in pH 6.0, for 60 s (association phase) and transferred to buffer-containing wells for additional 60 s (dissociation phase). Binding and dissociation were measured as changes over time in light interference after subtraction of parallel measurements from unloaded biosensors. Sensorgrams were fitted with a 1:1 binding model using the Octet data analysis software 8.1 (Fortebio, USA, 2015). All BLI experiments were repeated several times.

**Cells.** Vero E6 (ATCC® CRL-1586™) were obtained from the American Type Culture Collection. Cells were grown in Dulbecco's modified Eagle's medium (DMEM) supplemented with 10% fetal bovine serum (FBS), MEM non-essential amino acids (NEAA), 2 mM L-glutamine, 100 Units/ml penicillin, 0.1 mg/ml streptomycin and 12.5 Units/ml Nystatin (P/S/N) (Biological Industries, Israel). Cells were cultured at 37 °C, 5% CO₂ at 95% air atmosphere.

ExpiCHO-S (Thermoscientific, USA, Cat# A29127) were used for expression of recombinant proteins as described above.

**Plaque reduction neutralization test (PRNT).** Handling of SARS-CoV-2 was conducted in a BSL3 facility in accordance with the biosafety guidelines of the Israel Institute for Biological Research (IIBR). SARS-CoV-2 (GISAID accession EPI_ISL_406862) strain was kindly provided by Bundeswehr Institute of Microbiology, Munich, Germany. Virus stocks were propagated and tittered by infection of Vero E6 cells as recently described[44]. For PRNT, Vero E6 cells were plated

overnight (as detailed above) at a density of 5 × 10⁵ cells/well in 12-well plates. Ab samples were twofold serially diluted (ranging from 2 to 0.015 μg/ml) in 400 μl of MEM supplemented with 2% FBS, NEAA, 2 mM L-glutamine and P/S/N. Four hundred microliters containing 500 PFU/ml of SARS-CoV-2 virus were then added to the Ab solution supplemented with 0.25% guinea pig complement sera (Sigma, Israel) and the mixture was incubated at 37 °C, 5% CO₂ for 1 h. Monolayers were then washed once with DMEM w/o FBS and 200 μl of each Ab-virus mixture was added in triplicates to the cells for 1 h. Virus mixture w/o Ab was used as control. In all, 2 ml/well overlay [MEM containing 2% FBS and 0.4% Tragacanth (Sigma, Israel)] was added to each well and plates were further incubated at 37 °C, 5% CO₂ for 48 h. Following incubation, the overlay was aspirated and the cells were fixed and stained with 1 ml of crystal violet solution (Biological industries, Israel). The number of plaques in each well was scored and the NT₅₀ (Ab concentration at which the plaque number was reduced by 50%, compared to plaque number of the control, in the absence of Ab) was calculated using the Prism software version 8 (GraphPad Software Inc., USA).

**Animal experiments.** Treatment of animals was in accordance with regulations outlined in the U.S. Department of Agriculture (USDA) Animal Welfare Act and the conditions specified in the Guide for Care and Use of Laboratory Animals (National Institute of Health, 2011). Animal studies were approved by the local ethical committee on animal experiments (protocol number M-51-20 and M-56-20). Male and female K18-hACE2 transgenic (B6.CgTg(K18ACE2)2Prlmn/J HEMI) and C57BL/6 mice (Jackson Laboratories, USA) were maintained at 20–22 °C and a relative humidity of 50 ± 10% on a 12 h light/dark cycle, fed with commercial rodent chow (Koffolk Inc.), and provided with tap water ad libitum. The age of the animals at the time of the onset of experiments ranged between 6 and 10 weeks old. All animal experiments involving SARS-CoV-2 were conducted in a BSL3 facility.

Infection experiments were carried out using SARS-CoV-2, isolate Human 2019-nCoV ex China strain BavPat1/2020 that was kindly provided by Prof. Dr. Christian Drosten (Charité, Berlin, Germany) through the European Virus Archive —Global (EVAg Ref-SKU: 026V-03883). The original viral isolate was amplified by five passages and quantified by plaque titration assay in Vero E6 cells, and stored at −80 °C until use. The viral stock DNA sequence and coding capacity were confirmed as recently reported[45].

SARS-CoV-2 BavPat1/2020 virus diluted in PBS supplemented with 2% FBS (Biological Industries, Israel) was used to infect animals by intranasal instillation of anesthetized mice. For Abs protection evaluation, mice were treated with two doses (4 days apart) of 1 mg/mouse of MD65 Ab, first administered IP either 4 h prior, or 1–4 days post-infection with 200 PFU of SARS-CoV-2. Dose-dependency experiments were performed by single administration of 1, 0.1, and 0.01 mg/mouse (equivalent of 50, 5, and 0.5 mg/kg body weight, respectively) of the MD65 Ab 2 days following infection with 300 PFU SARS-CoV-2. Control groups were administered with PBS or isotype control Ab (anti-ricin MH75 Ab) at the indicated times. Body weight was monitored daily throughout the follow-up period post-infection. Mice were evaluated once a day for clinical signs of disease and dehydration. Euthanasia was applied when the animals exhibited irreversible veterinary-evaluated disease symptoms (rigidity, lack of any visible reaction to contact).

**Pharmacokinetics.** Pharmacokinetics was determined in 6- to 8-week-old male and female C57BL/6 mice following administration of 0.2 mg MD65 IgG-YTE Ab either by IV (n = 4; 200 μl of 1 mg/ml Ab) or IP [n = 5; 1 ml of 0.2 mg/ml Ab] route. At different time points, 5 μl blood samples were drawn from the tail vein, diluted 20-fold in PBS, and centrifuged for 10 min at 500g for the removal of red blood cells. Supernatant plasma fractions were stored at −20 °C until tested in ELISA for the Ab concentration determination as described above. Average Ab concentration at each time point was used for calculating the pharmacokinetic parameters using non-compartmental analysis (PK solutions 2.0; Summit Research Services, USA).

**Measurement of viral RNA by qRT-PCR.** Viral load in lungs of SARS-CoV-2-infected mice (200 PFU) treated with the MD65 Ab was quantified by qRT-PCR and by plaque assay (see above). Lungs were grinded in 1.5 ml PBS and 200 μl were added to LBF lysis buffer. RNA was extracted using RNAdvance Viral Kit on a Biomek i7 automated workstation (Beckman Coulter, IN) according to the manufacturer's protocol. Each sample was eluted in 50 μl of RNase-free water. RT-PCR was performed using the SensiFAST™ Probe Lo-ROX One-Step kit (Bioline, UK). Primers and probe sequences, targeting the SARS-CoV-2 E gene, were based on the Berlin protocol published in the WHO recommendation for the detection of SARS-CoV-2 [E_Sarbeco_F1 ACAGGTACGTTAATAGT TAATAGCGT, E_Sarbeco_R2 ATATTGCAGCAGTACGCACACA, E_Sarbeco_P1 ACACTAGCCATCCT-TACTGCGCTTCG]. The primers and probe were used at a final concentrations of 600 and 300 nM, respectively. The thermal cycling reaction was performed at 48 °C for 20 min for reverse transcription, followed by 95 °C for 2 min, and then 45 cycles of 15 s at 94 °C; 35 s at 60 °C for the E gene amplification. Cycle threshold (Ct)

values were converted to PFU equivalents (PFU Eqv.), according to a calibration curve determined in parallel.

**Lung histology**. Lungs were rapidly isolated, fixed in 4% PBS-buffered paraformaldehyde at room temperature for 1 week, followed by routine processing for paraffin embedding. Serial sections, 5-μm-thick, were cut and selected sections were stained with hematoxylin and eosin and examined by light microscopy. Camera images were taken using a Zeiss Axioskop microscope (Zeiss, Oberkochen, Germany) equipped with a Nikon DS-Ri1 camera controlled by a DS-U3 Digital Sight and the Nis-Elements-Br software suite (Nikon, Tokyo, Japan).

**Reporting summary**. Further information on research design is available in the Nature Research Reporting Summary linked to this article.

## Data availability
The MD65 antibody sequence has been reported in ref. [11]. The antibody is available (by contacting O.M. from the Israel Institute for Biological Research; ohadm@iibr.gov.il) for research purposes only under an MTA, which allows the use of the antibody for non-commercial purposes but not its disclosure to third parties. Source Data are provided with this paper. All other data are available from the corresponding author upon reasonable requests.

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

## Acknowledgements
We thank Prof. Dr. Christian Drosten at the Charité Universitätsmedizin, Institute of Virology, Berlin, Germany for providing the SARS-CoV-2 BavPat1/2020 strain. We wish

to express our gratitude to our colleagues Dr. Hagit Achdout, Dr. Nir Paran, Dr. Emanuelle Mamroud, Dr. Hadas Tamir, Dr. Yfat Yahalom-Ronen, Dr. Shay Weiss, Dr. Hadar Marcus, Dr. Noa Madar-Balakirski, Roy Avraham, Dr. Amir Ben-Shmuel, Dr. Amir Rosner, Dr. Tseela David, and Dr. Hani Dekel Jaoui for fruitful discussions and support. We would like to thank Moshe Mantzur, Tamar Aminov, Meni Girshengorn, and Sarah Borni for skillful and devoting technical assistance.

## Author contributions

R.R., T.N.-P., A.M., E.M., Y.L., R.A., R.F., M.A., E.E., D.G., E.B.V., S.M., B.P., A.Z., S.L., A.B.-D., Y.E., and O.M. designed, carried out, and analyzed the data. T.C., S.Y., S.C.S., and T.I. added fruitful discussions, reviewed, and edited the manuscript. R.R. and O.M. supervised the project. All authors have approved the final manuscript.

## Competing interests

Patent application for the described antibody was filed by the Israel Institute for Biological Research. None of the authors declared any additional competing interests.
