## [Peer Review File · Nature Communications]

REVIEWER COMMENTS

Reviewer #1 (Remarks to the Author):

The manuscript entitled 'Post-exposure protection of SARS-CoV-2 lethal infected K18 hACE2 mice transgenic mice by neutralizing human monoclonal antibody by Rosenfeld et al. describes the prophylactic and therapeutic effects of a specific human monoclonal antibody and modified variants to against SARS-CoV-2 infection in vivo using the now well established K18 hACE2 mouse model. There are several major concerns that authors need to address before this study can be suitable for publication.

Major concerns:

1/ In vivo experiments using K18 hACE2 mice infected with what is considered a low dose (2×10^2) and treated with what is considered a high dose of mAbs (around approx. 70 mg/Kg of weight considering two doses and a mouse averaging 28 g of weight).

Authors need to justify why they infected with such a low SARS-CoV-2 inoculum (2×10^2) and treated with such a large dose of neutralizing antibody. This low inoculum using other SARS-CoV-2 strains is shown to be non-lethal. If the purpose of this study is to evaluate the efficacy of these neutralizing antibodies against SARS-CoV-2 (strain independent), then it is important to perform a dose titration study in vivo (this is in the methods sections but not data are presented and/or discussed), as well as perform this study with both a low (1×10^2 - 1×10^3) and a high (1×10^5) SARS-CoV-2 inoculum to verify the efficacy of the antibodies at different concentrations.

2/ Specify why 1 mg per mouse was used. Normally administration of Ab are standardized by weight of the animal. This is particularly important in the injection at 4-DPI. The amount of 1 mg/mouse/twice is considered a high concentration of mAbs (approx. 72 mg/Kg per mouse, considering the average weight of 28 g per mouse).

3/ Importantly, the impact of neutralizing Abs on SARS-CoV-2 viral titers in the lungs and brain needs to be reported.

4/ Report of the cytotoxicity of the neutralizing Abs (tissue histopathology/inflammation) treatment is required to confirm the results observed, especially if the high SARS-CoV-2 inoculum will be used to confirm that these mAbs at different concentrations perform well.

5/ Weight in the morbidity curves for mice receiving MD65 by 3-DPI are not reported after 6-DPI, even so 60% of the animals survived. It will be important to report if the 60% survivors reestablished their body weight, even when the treatment is delayed by 3-DPI.

6/ Controls as reported in Figure 4. The same results obtained from control mice in the prophylaxis study were plotted for the therapeutic studies, but mice in the therapeutic studies were treated differently than in the prophylaxis study. Although in this case it may be expected similar results, authors need to justify or address why this decision was made.

7/ Provide detailed information about how the SARS-CoV-2 virus strain was handled, how many passages and how the viral passage used was verified for mutations/deletions that could affect its infectivity/virulence.

8/ Verify that mouse data are compiled from 4 different experiments (as reported) and if the 'n' values are compiled or are the ones used in each experiment. If 'n' values are compiled, revise them, as it seems that a very low number of mice (n=1-2?) were used in each experiment. Clarify this point to avoid confusions.

9/ Authors mentioned the use of females and males but there is not an intent of presenting data based on sex to determine if sex plays a role on the responses observed.

Minor concerns:

1/ Authors show a list of references containing both deposit papers in BioRxiv (which is OK) and published. Revise the literature and provide a complete report of the studies being performed validating the K18 hACE2 mouse as a model for SARS-CoV-2 infection.

2/ Justify the in vitro binding assays performed at pH 6.0, when the pH of blood is 7.35 to 7.45.

3/ In Figure 1c is difficult to clearly discern the lines. It is not sure if the lines are perfectly overlapping or data are missing. If overlapping, indicate this in the figure legend.

4/ Lines 127-129: The concentrations reported do not seem to be the same, these can be similar but not the same. In addition, Fig 1f does not seem to match its description in the text, e.g. reaching C_{max} of 40 micrograms per ml within 180 min.

5/ Lines 154-156: A massive weight loss is mentioned but mice only lost 5% body weight. Define what the authors understand for a massive weight loss.

6/ Line 156: 75% mortality is observed by 8-DPI and not by 6-DPI as reported in the text.

7/ Scheme in Fig 3 is confusing. Remove the 4 h prior infection from the figure or add this text only under the first green arrow, as leads to confusion as it is. In this figure the different tonalities of green color are difficult to discern, especially between MD65 3 DPI and 4 DPI.

8/ Line 237: Reference formatting.

9/ It is not clear why the methods section describes the use of C57BL/6 mice as well as pharmacokinetics in vivo, as these do not appear to be reported in any way in the manuscript.

Reviewer #2 (Remarks to the Author):

In the study by Rosenfeld et al the ability of a mAb MD65 originally isolated by phage display was tested for the ability to prevent or treat SARS-CoV infection when delivered passively.

The mAb, originally isolated using phage display, which neutralizes SARS-CoV-2 by blocking the spike-ACE2 interaction was converted from an scFV format to an IgG, and then further modified to have high affinity for neonatal Fc receptors. Although this would presumably extend the half life in humans (or NHPs) only the in vitro binding properties are reported here.

The authors perform a pharmacokinetic study in mice using both IV and IP administration to determine the half life of the mAb.

The mAb is then assessed for its ability to protect against and treat established SARS-CoV2 infection in mice expressing ACE2 under the control of the K18 promoter. At the viral dose given most of the untreated mice succumb to death within 6-9 days post infection.

Given prophylactically the mAb protected against weight loss and death. When given after viral challenge the mAb prevented death and weight loss if administered before 4 days post-infection.

The authors further demonstrate that the mAb given as treatment, does not prevent the development of de novo antibody responses.

The study is well written and for the most part, the results are clearly laid out.

My major concern is that this model may not be predictive of mAb efficacy in humans. Most people do not experience severe disease and it can take days to weeks to develop, whereas the majority of the mice in the study die within a week. So its difficult to envision how the findings of this study would translate since the efficacy of the mAb falls off significantly if given after 4 days of infection.

It's not clear from the methods if death is an endpoint in the study or whether the animals are euthanized after 20% weight-loss. If it's the latter, would the infected animals recover (like Syrian hamsters) if they were left alone.

Necropsy was not performed on the animals to evaluate lung pathology or viral loads, so it can't be concluded that the mAb provided "full protection" when given prophylactically as claimed.

I'd suggest changing the color scheme for some of the figures. Eg it's hard to tell which group is which in 3e.

Detailed list of responses and amendments introduced in the revised version

Comments are recapitulated below (in black) followed by our responses (in red)

Reviewer #1 (Remarks to the Author):

The manuscript entitled 'Post-exposure protection of SARS-CoV-2 lethal infected K18 hACE2 mice transgenic mice by neutralizing human monoclonal antibody by Rosenfeld et al. describes the prophylactic and therapeutic effects of a specific human monoclonal antibody and modified variants to against SARS-CoV-2 infection in vivo using the now well established K18 hACE2 mouse model. There are several major concerns that authors need to address before this study can be suitable for publication.

Major concerns

1. In vivo experiments using K18 hACE2 mice infected with what is considered a low dose (2×10^2) and treated with what is considered a high dose of mAbs (around approx. 70 mg/Kg of weight considering two doses and a mouse averaging 28 g of weight).

Authors need to justify why they infected with such a low SARS-CoV-2 inoculum (2×10^2) and treated with such a large dose of neutralizing antibody. This low inoculum using other SARS-CoV-2 strains is shown to be non-lethal. If the purpose of this study is to evaluate the efficacy of these neutralizing antibodies against SARS-CoV-2 (strain independent), then it is important to perform a dose titration study in vivo (this is in the methods sections but not data are presented and/or discussed), as well as perform this study with both a low (1×10^2 - 1×10^3) and a high (1×10^5) SARS-CoV-2 inoculum to verify the efficacy of the antibodies at different concentrations.

The low dose of SARS-CoV-2 infection employed in the study is due to the high lethality of the virus preparation used (the 200 PFU dose employed was found to be consistently associated with 80% mortality). This high lethality of the virus is probably strain and handling specific: the strain employed here is the SARS-CoV-2 BavPat1/2020 strain which was stored at -80°C and passaged only 5 times from original isolation, therefore preserving its virulence. Additionally, the viral stock DNA sequence and coding capacity were confirmed. The furin cleavage site, essential for high virulence of SARS CoV-2, was retained. This information is now further clarified in the revised manuscript (lines 456-458 and new reference 45; in response to this comment as well as comment 7, see below).

Yet, to meet the reviewer comment, and to demonstrate that the efficient neutralization promoted by the antibody does not reflect an unusual high antibody/virion ratio, we repeated the experiment with much lower amounts of antibody (up to 200 times lower dose). The results of this novel independent experiment establishing a dose-dependency of the treatment and the ability of the antibody to protect the mice at a much lower dose, are incorporated in the new Figure 6 of the revised manuscript. Furthermore, this dose-dependent experiment was performed by administrating the antibody 48 hours after the viral infection, a stage at which the viral load in the lungs is considerably higher than the input dose.

2. Specify why 1 mg per mouse was used. Normally administration of Ab are standardized by weight of the animal. This is particularly important in the injection at 4-DPI. The amount of 1 mg/mouse/twice is considered a high concentration of mAbs (approx. 72 mg/Kg per mouse, considering the average weight of 28 g per mouse).

We indeed used initially, as described, a double dose of 1 mg antibody/mouse, to evaluate the protective ability of the antibody. This dosage is explicitly rationalized in the text which reads

“...In order to maintain a steady antibody levels in the circulation for about 8 days [commensurate with the 7.5 days mean time to death (MTTD) in the control group], and taking into consideration the pharmacokinetic profile, the antibody was administered via the IP route in two successive treatments, 4 days apart.”

Yet, as requested in this comment, data pertaining to efficiency of much lower doses of antibody that were given only once are included in the revised manuscript (see new Figure 6 and response to comment 1 above). This new experiment employing single doses of 1, 0.1 and 0.01 mg Ab/animal (equivalent of 50, 5, and 0.5 mg/Kg body weight, respectively) 2 post infection demonstrated that a dose 20 times lower than the initial treatment preserved ability to fully protect the animals. A most significant extension of the time-to-death was promoted by antibody doses as low as 1/200 of the initial dose. The new experiment resulted in the addition of a new sub-section in the “Results” and appropriate clarifications in the “Methods” (lines 291-304 and 463-466).

3. Importantly, the impact of neutralizing Abs on SARS-CoV-2 viral titers in the lungs and brain needs to be reported.

As requested by the reviewer, we have performed a novel series of experiments to evaluate the viral titers in the lungs of antibody-treated mice versus the control untreated infected animals (new Figure 4 in the revised manuscript). Although SARS-CoV-2 was suggested to be present in other tissues of K18 mice (including brain), lungs represent the main organ where infective multiplying viruses are detected (see for example Yinda CK et al., 2020; Winkler ES et al., 2020). The added data were obtained in experiments in which, the viral load was determined in lungs by two alternative yet complementary assays (RT-qPCR and plaque assay). Incorporation of these new data resulted also in addition of one new sub-section in the Results (lines 207-222) as well as in the Methods (lines 484-500). We agree with the reviewer that this information is important substantiating the conclusion of the study and therefore a brief sentence was also added to the abstract (lines 31-33).

4. Report of the cytotoxicity of the neutralizing Abs (tissue histopathology/inflammation) treatment is required to confirm the results observed, especially if the high SARS-CoV-2 inoculum will be used to confirm that these mAbs at different concentrations perform well.

As requested, we now added histopathology data to evaluate the inflammatory status of the lungs of mice that were infected with SARS CoV-2 and treated with MD65 antibody, 2 days post-infection. (new Figure 4, lines 224-246 and 502-509 and Supplementary Figure 3).

5. Weight in the morbidity curves for mice receiving MD65 by 3-DPI are not reported after 6-DPI, even so 60% of the animals survived. It will be important to report if the 60% survivors reestablished their body weight, even when the treatment is delayed by 3-DPI.

Indeed, surviving mice that were treated with MD65 4-days post infection, quickly recovered by gaining weight. The requested information was incorporated in the modified Figure 3 (see panel e) of the revised manuscript and a clarification was included in the legend of Fig. 3 (lines 187-205) as well as in the text (lines 180-182).

6. Controls as reported in Figure 3. The same results obtained from control mice in the prophylaxis study were plotted for the therapeutic studies, but mice in the therapeutic studies were treated differently than in the prophylaxis study. Although in this case it may be expected similar results, authors need to justify or address why this decision was made.

A new version of Fig. 3 was included in the revised manuscript (see also response to minor comment 7). In this novel version of the figure, the parameters (body weight and survival) of control animals pertaining to the specific experiments depicted in each panel, are shown. Clarifications regarding the control animals were also included in the legend to Fig. 3.

7. Provide detailed information about how the SARS-CoV-2 virus strain was handled, how many passages and how the viral passage used was verified for mutations/deletions that could affect its infectivity/virulence.

As mentioned above, the strain employed here is the SARS-CoV-2 BavPat1/2020 strain which was stored at -80°C and passaged only 5 times from original isolation, therefore preserving its virulence. Additionally, the viral stock DNA sequence was confirmed. The requested details were included in the “Methods” section of the revised manuscript (lines 456-458).

8. Verify that mouse data are compiled from 4 different experiments (as reported) and if the ‘n’ values are compiled or are the ones used in each experiment. If ‘n’ values are compiled, revise them, as it seems that a very low number of mice ($n=1-2?$) were used in each experiment. Clarify this point to avoid confusions.

For clarification, a completely modified Fig. 3 replaces the original one in the revised manuscript. The n numbers refer to the number of animals within one experimental group in one iteration of an experiment. Exception to this is in panel e and f, in which the groups described as $n>5$ refer to combined results obtained from several experiments, each one including at least 4 animals. This issue is clarified in the legend to Fig. 3 (lines 187-205).

9. Authors mentioned the use of females and males but there is not an intent of presenting data based on sex to determine if sex plays a role on the responses observed.

A sentence addressing this issue was inserted in the text of the revised manuscript (lines 156-159). As indicated, some of the experiments included groups of male and female animals. This is now exemplified in the novel Supplementary Figure 1 annexed to the revised manuscript. Indeed, no differences were detected between male and female mice, as anticipated. Yet we do acknowledge that this issue deserves further confirmation by a larger cohort of animals. This is also indicated in the revised manuscript.

Minor concerns

1. Authors show a list of references containing both deposit papers in BioRxiv (which is OK) and published. Revise the literature and provide a complete report of the studies being performed validating the K18 hACE2 mouse as a model for SARS-CoV-2 infection.

Indeed 3 important articles describing the K18 model were published that were omitted inadvertently in the initial version of the manuscript. The revised version includes specific mentions to these three studies in several locations which are cited now in the reference section as references 28, 29 and 40.

2. Justify the in vitro binding assays performed at pH 6.0, when the pH of blood is 7.35 to 7.45.

The YTE mutation induces higher affinity of the Fc to the FcRn only in acidic pH, resulting in improved protection of the antibody from lysosomal degradation. At neutral pH, the YTE mutation does not affect the binding to the FcRn, enabling dissociation and release of the

antibody to the bloodstream. Accordingly, and as widely accepted and reported for other antibodies in the YTE format, the affinity of the antibody to the FcRn is measured at pH 6.0.

3. In Figure 1c is difficult to clearly discern the lines. It is not sure if the lines are perfectly overlapping or data are missing. If overlapping, indicate this in the figure legend.

Indeed, the pink and the cyan curves overlap in Fig. 1c. A clarification was introduced into the legend (lines 98-99), as requested.

4. Lines 127-129: The concentrations reported do not seem to be the same, these can be similar but not the same. In addition, Fig 1f does not seem to match its description in the text, e.g. reaching C_{max} of 40 micrograms per ml within 180 min.

The paragraph was modified and the information is now clarified (lines 128-129). The original word “similar” was replaced by “commensurate”. Furthermore, the raw data are now provided in the “Source Data” on-line section of the manuscript.

5. Lines 154-156: A massive weight loss is mentioned but mice only lost 5% body weight. Define what the authors understand for a massive weight loss.

The word “massive” was replaced by “significant” (line 155 in the revised manuscript).

6. Line 156: 75% mortality is observed by 8-DPI and not by 6-DPI as reported in the text.

The text was modified as requested (new lines 155-159). Please note that Fig. 3 and the corresponding text and legends were modified in the revised manuscript.

7. Scheme in Fig 3 is confusing. Remove the 4 h prior infection from the figure or add this text only under the first green arrow, as leads to confusion as it is. In this figure the different tonalities of green color are difficult to discern, especially between MD65 3 DPI and 4 DPI.

A completely modified Fig. 3 (including its legend) replaces the original one in the revised manuscript. Note that Fig. 3 is now divided into 2 clearly distinct parts pertaining to prophylactic (left) and post-exposure (right) experiments. The novel Fig. 3 incorporates a different partition into panels as well as modified, less confusing colors. We believe that the novel schemes in the revised figure are considerably easier to comprehend.

8. Line 237: Reference formatting.

This reference formatting was corrected in the revised manuscript.

9. It is not clear why the methods section describes the use of C57BL/6 mice as well as pharmacokinetics in vivo, as these do not appear to be reported in any way in the manuscript.

Indeed, as mentioned in the method section, the PK experiments were performed using C57BL/6 mice (the parental strain of the K18-hACE2 transgenic mice). Fig 2 legend as well as the PK experiment description in Methods section, were corrected accordingly.

Reviewer #2 (Remarks to the Author):

In the study by Rosenfeld et al the ability of a mAb MD65 originally isolated by phage display was tested for the ability to prevent or treat SARS-CoV infection when delivered passively.

The mAb, originally isolated using phage display, which neutralizes SARS-CoV-2 by blocking the spike-ACE2 interaction was converted from an scFV format to an IgG, and then further modified to have high affinity for neonatal Fc receptors. Although this would presumably extend the half life in humans (or NHPs) only the in vitro binding properties are reported here.

The authors perform a pharmacokinetic study in mice using both IV and IP administration to determine the half life of the mAb.

The mAb is then assessed for its ability to protect against and treat established SARS-CoV2 infection in mice expressing ACE2 under the control of the K18 promoter. At the viral dose given most of the untreated mice succumb to death within 6-9 days post infection.

Given prophylactically the mAb protected against weight loss and death. When given after viral challenge the mAb prevented death and weight loss if administered before 4 days post-infection.

The authors further demonstrate that the mAb given as treatment, does not prevent the development of de novo antibody responses.

The study is well written and for the most part, the results are clearly laid out.

My major concern is that this model may not be predictive of mAb efficacy in humans. Most people do not experience severe disease and it can take days to weeks to develop, whereas the majority of the mice in the study die within a week. So its difficult to envision how the findings of this study would translate since the efficacy of the mAb falls off significantly if given after 4 days of infection.

We are aware of the fact that the K18-hACE2 mice exhibit enhanced sensitivity to SARS-Cov-2 infection compared to the evolution of the disease in humans. Yet, this model is widely accepted and highly appropriate for assessment of the efficiency of counter-measures in general and of immunotherapies in particular. Due to their enhanced sensitivity, they impose a higher stringency on the efficiency of the treatment under study. Moreover, the current preclinical study represents a first feasibility demonstration of the protective potential of the MD65 antibody, and the extrapolation of the findings documented here for implementation of the antibodies as a bona-fide therapeutic product for human use will require additional studies. This aspect is strongly suggested in the concluding paragraph of the manuscript.

Please note that the revised version of the manuscript was extensively modified and now includes additional data demonstrating the efficacy of the MD65 antibody therapy. The novel data describe the dose dependency of the treatment, the impact of the treatment on the viral burden in the lungs of infected animals, as well as the physiological state of the lung tissues following the diseases.

It's not clear from the methods if death is an endpoint in the study or whether the animals are euthanized after 20% weight-loss. If it's the latter, would the infected animals recover (like Syrian hamsters) if they were left alone.

Indeed, body weight loss was not considered a parameter for end point and accordingly the endpoint was applied when the animals exhibited additional terminal symptoms (rigidity, lack of any visible reaction to contact). Euthanasia was performed only in cases of irreversible disease. A clarification was introduced in the revised manuscript (lines 469-471).

Necropsy was not performed on the animals to evaluate lung pathology or viral loads, so it can't be concluded that the mAb provided "full protection" when given prophylactically as claimed.

To meet the reviewer comment, two additional set of experiments were performed: 1. Measurement of viral load in the lungs of Ab-treated versus untreated mice; 2. Histopathological evaluation of the Ab-treated animals. The revised manuscript includes now new data pertaining to lung pathology and viral loads (see new Fig. 4 and Supplementary Figure 3 in the revised manuscript).

I'd suggest changing the color scheme for some of the figures. Eg it's hard to tell which group is which in 3e.

The colors were modified as suggested.

REVIEWERS' COMMENTS

Reviewer #1 (Remarks to the Author):

This study improved, but authors still have some claims that may need revision. There are other studies, e.g. Rogers et al. 2020 and Baum et al. 2020, both in Science, in particular the latter one, showing the efficiency of a cocktail of neutralizing antibodies against SARS-CoV-2 infection in both non-human primates and hamsters; thus, authors may need to rephrase their last paragraph where they claim that it is the first study showing the use the mAbs as therapeutic treatment against SARS-CoV-2 infection.

Reviewer #2 (Remarks to the Author):

The authors have addressed all my concerns in the revised manuscript and I believe it to be significantly improved.

Manuscript #NCOMMS-20-41900A, final revision

Response and amendments introduced in the revised version

Comments are recapitulated below (in black) followed by our responses (in red)

Reviewer #1 (Remarks to the Author):

This study improved, but authors still have some claims that may need revision. There are other studies, e.g. Rogers et al. 2020 and Baum et al. 2020, both in Science, in particular the latter one, showing the efficiency of a cocktail of neutralizing antibodies against SARS-CoV-2 infection in both non-human primates and hamsters; thus, authors may need to rephrase their last paragraph where they claim that it is the first study showing the use the mAbs as therapeutic treatment against SARS-CoV-2 infection.

In response to the reviewer comment, we have now rephrased the concluding paragraph and included the two suggested references, which in the original manuscript were only invoked in the introduction section.

Reviewer #2 (Remarks to the Author):

The authors have addressed all my concerns in the revised manuscript and I believe it to be significantly improved.

Thank you.